# Altered Thermal Behavior of Blood Plasma Proteome Related to Inflammatory Cytokines in Early Pregnancy Loss

**DOI:** 10.3390/ijms23158764

**Published:** 2022-08-06

**Authors:** Regina Komsa-Penkova, Avgustina Danailova, Sashka Krumova, Galya Georgieva, Ina Giosheva, Lidia Gartcheva, Ivan Iliev, Emil Gartchev, Kameliya Kercheva, Alexey Savov, Svetla Todinova

**Affiliations:** 1Department of Biochemistry, Medical University—Pleven, Sv. Kliment Ohridski Str. 1, 5800 Pleven, Bulgaria; 2Institute of Biophysics and Biomedical Engineering, Bulgarian Academy of Sciences, “Acad. G. Bontchev” Str. 21, 1113 Sofia, Bulgaria; 3University Hospital of Obstetrics and Gynecology “Maichin Dom”, Medical University Sofia, Zdrave Str. 2, 1431 Sofia, Bulgaria; 4National Specialized Hospital for Active Treating of Haematological Diseases, 1756 Sofia, Bulgaria; 5Institute of Experimental Morphology, Pathology and Anthropology with Museum, Bulgarian Academy of Sciences, Acad. G. Bonchev Str., Bl. 25, 1113 Sofia, Bulgaria

**Keywords:** blood plasma proteome, early pregnancy loss, differential scanning calorimetry, 4G/5G polymorphism in *PAI-1* gene, tumor necrosis factor α, interleukin-6

## Abstract

Early pregnancy loss (EPL) is a relatively common pathology of which almost 50% of cases remain idiopathic. In the search for novel biomarkers, differential scanning calorimetry (DSC) is intensively used to characterize the thermodynamic behavior of blood plasma/serum proteome in health and disease. Herein, for the first time, we investigate the DSC denaturation profiles of blood plasma derived from patients suffering EPL compared to healthy pregnant and non-pregnant women. Data analysis reveals that 58% of the EPL thermograms differ significantly from those of healthy pregnant women. Thermal stabilization of a fraction of albumin-assigned transition with concomitant suppression of the major and enhancement of the globulin-assigned transition are characteristic features of EPL calorimetric profiles that could be used as a new indicator of a risk pregnancy. The presented results suggest an altered composition or intermolecular interactions of the plasma proteome of women with EPL. In addition, the alterations of the EPL thermograms correlate with the increased blood levels of tumor necrosis factor-α (TNF-α) and interleukin-6 (IL-6) and a higher prevalence of the polymorphism in the plasminogen activator inhibitor type-1 (*PAI-1)* gene, suggesting an expression of an overall enhanced immune response. The concomitant changes in plasma thermograms confirm the potential of the DSC approach for distinguishing changes in the pathological state of the blood plasma proteome.

## 1. Introduction

Early pregnancy loss (EPL) or fetal loss is a relatively common pathology [1]. Some of the causes of fetal loss include chromosomal abnormalities [2,3], genetic syndromes, thrombophilia [4,5], infections [6], antiphospholipid syndrome, and maternal diseases such as diabetes and hypertension, and inflammation [7]. Nevertheless, more than 50% of EPLs remain idiopathic [8]. Despite a wide range of investigations, no specific biomarkers with a high predictive value of threatening pregnancy losses have been identified.

Normal pregnancy is a complex and dynamic process during which the immune system balances the mother’s physiological changes and the growing needs of the fetus. During pregnancy, the function of the maternal immune system is to protect the mother and the fetus from pathogens, as well as prevent the rejection of the allogeneic transplant of the fetal–placental unit [9]. On the surface of trophoblast cells MHC class, I antigens are presented by antigen-presenting cells (APCs) to maternal helper T cells [10]. Thus, activated lymphocytes start to secrete pro- and anti-inflammatory cytokines. Th1 secretion includes interferon-gamma (INF-γ), interleukin Il-1β (Il-1β), Il-2, Il-12, Il-15, and tumor necrosis factor α (TNF-α) [11], whereas Th2 cells produce mainly Il-10, as well as Il-4, Il-5, Il-6, Il-9, Il-13, tumor growth factor (TGF-β1). Th1 cytokines participate in the late hypersensitivity reaction and direct the cellular response, while Th2 cytokines activate B cells, thereby enhancing the humoral response [12]. Circulating cytokine levels in the mother’s bloodstream reflect her immune status [13]. IL-6 and TNF-α are widely expressed in the female reproductive tract and gestational tissues and exert regulatory functions in embryo implantation and placental development [14,15]. IL-6 and TNF-α play a significant role in many pregnancy complications, including early miscarriage suggesting their pathogenesis in such conditions [16,17].

Embryo implantation and trophoblast invasion involve a local inflammatory environment that supports blastocyst adhesion, a consequent invasion of the blastocyst and the trophoblast, and tissue reorganization in the uterine wall [18]. The embryo, therefore, develops in a low-oxygen environment, thus protecting differentiating cells from damaging reactive oxygen species (ROS). An over-expressed inflammatory reaction and ROS production can lead to pregnancy complications or early miscarriage [19]. The invasion of the trophoblast could be severely impaired due to the high levels of oxidative stress in the periphery [20]. Hence, extensive syncytiotrophoblastic oxidative damage is likely one of the major factors of EPL. 

The discovery of new biomarkers in biological fluids is one of the still unresolved challenges that researchers are facing due to the high molecular components’ complexity, as well as their wide and dynamic concentration range in the bloodstream. The dramatic changes that occur in the mother’s body also exert an effect on the blood plasma proteome from the beginning of pregnancy to birth [21]. For example, it is known that plasma proteins, such as free protein S and protein C, responsible for the reduction of fibrinolytic activity, take on lower levels during pregnancy [22]. Polymorphisms associated with these proteins can cause severe complications during pregnancy such as first trimester miscarriages. In line with this, Trauscht-Van Horn et al. found a relationship between protein C deficiency, miscarriage, and thrombosis [23,24]. Fibrinolytic activity is also reduced in normal pregnancy due to the increased levels of plasminogen activator inhibitor type 1 (*PAI-1*), which increases fivefold compared to the non-pregnant state, as well as the plasminogen activator inhibitor derived from placenta type 2 (*PAI-2*) [25,26]. However, overexpression of *PAI-1* in the blood is associated with an augmented risk for worse pregnancy outcomes [26]. Many studies have been focused on establishing the role of pregnancy-specific proteins, such as pregnancy zone protein, pregnancy-associated plasma protein-A (PAPP-A), placental pregnancy-specific β1-glycoprotein (SP1), inhibin A [27], activin A, pregnancy-associated-α2-globulin and sE-selectin in normal pregnancies and determine their significance in pathologies [28,29,30]. However, these studies often give false-positive results and therefore should be combined with other methods.

Blood plasma is a potential source of protein biomarkers for the discovery of many pathologies. Most of the studies are focused on the presence, absence, or relative abundance of specific proteins, which are known to be related to the disease. Nevertheless, it is also of great importance to study the deviations in their molecular interactions, i.e., the formation of different macromolecular complexes, and/or ligand binding.

In recent years, differential scanning calorimetry (DSC) is used intensively to characterize the thermodynamic behavior of blood plasma/serum proteome in health and disease. The high sensitivity of the DSC method to the change, both in the level and the interactions of plasma proteins (with emphasis on the major plasma proteins), makes it suitable for the study of this complex body fluid [31]. The DSC approach is currently mainly used to identify calorimetric markers for cancer [32,33,34,35,36,37,38,39], however, it has been applied in a few studies to detect abnormalities in high-risk pregnancies [40,41].

In the present work, the DSC approach is used to study the denaturation profiles of blood plasma derived from EPL patients. We reveal that 58% of the studied profiles differ significantly from those of non-pregnant and pregnant controls in the shape of the thermogram and in the denaturation temperatures and excess heat capacities of the major thermal transitions.

## 2. Results

### 2.1. Patient Characteristics

We compared the main clinical and biochemical parameters of blood plasma from volunteer women with EPL, with those of gestationally and age-matched pregnant controls (PC1, where 1 stands for the first trimester) and healthy non-pregnant women (NPC). The main characteristics (age, BMI, and the gestational week (GW) at the time of abortion) of the studied groups and the blood parameters (total protein (TP); human serum albumin (HSA); C-reactive protein (CRP), and fibrinogen (Fg) concentration) are presented in Table 1.

It was found that the values of the basic biochemical parameters of the EPL group did not differ significantly from those of the two control groups. The only exception was the level of CRP protein for EPL patients which was statistically lower than that of the PC1 group and similar to the NPC value.

### 2.2. Calorimetric Profiles of Blood Plasma Derived from Healthy Non-Pregnant and Pregnant Women

The average thermogram obtained from the DSC curves of blood plasma from 18 non-pregnant women (NPC group, Figure 1) was similar to the one previously published by us for a heterogeneous group of individuals of both sexes [36]. The denaturation transitions of Fg (≈50 °C), HSA (≈62 °C), and immunoglobulins (Igs ≈ 68 °C) were clearly distinguished, and the ratio of the specific heat capacities of HSA and immunoglobulins (Ig)s assigned transitions, c_P_^HSA^/c_P_^Igs^, was 2.0 ± 0.3 (Table 2). The high-temperature region (above 70 °C) was characterized by two shoulders—at about 75 °C and 82 °C, which can be attributed to the denaturation of complement C3 proteins and transferrin and IgG, respectively, as already described in Garbett et al. [42].

The plasma DSC profiles of the PC1 group were very similar to those of the NPC group, but a slightly higher specific heat capacity of the transitions assigned to fibrinogen and albumin denaturation was observed (Figure 1, Table 2). A statistical difference compared to NPC was found only for the first transition (*p* < 0.05). The denaturation transitions of complement C3 proteins (at about 75 °C) and transferrin and IgG (at 82 °C) were not clearly visible for PC1 samples. The calorimetric enthalpy (∆H) and the weighted average center of the thermogram (T_FM_) for PC1 samples also did not vary significantly from those of the NPC group.

### 2.3. Calorimetric Profiles of Blood Plasma Derived from Patients with Early Pregnancy Loss

In order to obtain reliable and statistically correct data, we compared the calorimetric curves of blood plasma from women with EPL with those of pregnant controls in the first trimester of pregnancy. Depending on the main calorimetric characteristics, i.e., the midpoint temperatures, and the respective heat capacities of the HSA and Igs assigned transitions, as well as their ratio c_P_^HSA^/c_P_^Igs^, the plasma thermograms of the EPL patients, were classified into two groups, designated as EPL1 and EPL2. To quantify the similarities/dissimilarities between the patients’ thermograms and those of the respective pregnant controls the statistical methodology developed by Fish et al. [43] was applied. The similarity metric ρ, which combines two factors: similarity in shape (Pearson’s correlation coefficient, r) and in space (spatial distance metric P) was determined for each patient’s calorimetric profile in relation to a set of control thermograms.

The EPL1 set included 42% of the studied patients’ cases. It was established that EPL1 thermograms differed from PC1 (Figure 2A) in terms of the lower amplitude of fibrinogen and albumin assigned transition (*p* < 0.05, Table 3). However, the shape and the midpoint transitions’ temperatures did not vary from the PC1 (*p* > 0.05) and consequently, the similarity metric ρ had a high value (Table 3).

The DSC profiles of the EPL2 group, which consisted of 46% of EPL cases, clearly differed from the respective controls (Figure 2B). The main transition (T_m_^HSA^) was upshifted by 2 °C accompanied by a considerably reduced c_P_^HSA^, compared to that of the PC1. Another characteristic feature of EPL2 profiles was the presence of a transition at about 60 °C and increased amplitude of Igs transition (Figure 2B). All these changes lead to a significant reduction in the c_P_^HSA^/c_P_^Igs^ ratio, increased calorimetric enthalpy ΔH and shifting of T_FM_ towards higher temperatures by nearly 3 °C (Table 3).

Three of the recorded EPL calorimetric curves (cases 4, 10, and 12, designated as EPL_case4_, EPL_case10_, and EPL_case12_, respectively, (Figure 2C)) demonstrated specific characteristics that distinguish them from the already defined EPL groups. The main characteristic of the EPL_case4_ DSC curve was the destabilization of albumin transition (by more than 3 °C), along with a reduction in c_P_^Igs^ and c_P_^HSA^/c_P_^Igs^ values compared to the PC1 group (Table 3). The blood plasma of the other two cases (EPL_case10_ and EPL_case12_) showed drastically different thermodynamic behavior. For both cases, we observed strong stabilization of the main transitions, as well as the one of Fg, by more than 2 °C. An additional feature of cases 10 and 12 that distinguishes them from all the other groups are the clearly pronounced transition at 82 °C and 86 °C, respectively. T_FM_ for these two cases had the highest registered value, which clearly indicates overall stabilization of the plasma proteome. The metric similarity of the ungrouped thermograms compared to the reference PC1 group varied within the 0.52 < ρ < 0.81 range (Table 3).

### 2.4. Plasma Protein Fractions of Control and EPL Samples

The data obtained by capillary electrophoresis are given in Table 4. The results analysis shows that the α2-globulin fraction (comprised of ceruloplasmin, α2-macroglobulin, and haptoglobin) for PC1 samples is above the reference values. The α1-globulin fraction (α1-antitrypsin, thyroid-binding globulin, and transcortin contribute to this band) for all patients’ groups was enlarged compared to the reference values and to that for the PC1 set, while the β2-globulin fraction (composed mainly of complement C3 and IgA proteins) was increased for EPL2 and for the two ungrouped cases EPL_case10_ and EPL_case12_ (Table 4). Statistically different values were also found for the β1-globulin fraction (composed mostly of transferrin) in cases EPL_case10_ and EPL_case12_, compared to the reference values and to the other groups under study. For all other plasma protein fractions, the values were within the reference range.

### 2.5. Carriage of Thrombophilia Polymorphism in 675 4G/4G in the PAI-1 Thrombophilia Gene

We determined the polymorphism of 675 4G/5G in the *PAI-1* thrombophilia gene for each of the participants in this study. We found that the incidence of carriage of *PAI-1* polymorphism in the EPL2 group was significantly higher (*p* < 0.05) as compared to healthy controls (NPC + PC) (19.2%) (Table 5). It is to be noted that EPL_case10_ and EPL_case12_ were both found to be carriers of 4G/4G polymorphism in the *PAI-1* gene, whereas EPL_case4_ was a carrier of wild type.

### 2.6. Blood Plasma Cytokine Levels (TNF-α and IL-6) of Control and EPL Samples

ELISA testing showed increased levels of TNF-α for the pregnant controls (28 ± 5 pg/mL, *p* < 0.05), compared to the non-pregnant control (2.7 ± 1.1 pg/mL) group. The levels of TNF-α for the EPL1 subset (27.3 ± 6.2 pg/mL, *p* > 0.05) were similar to the corresponding PC1 group, while that for the EPL2 was almost twice higher values (*p* < 0.05) than those of EPL1 (Figure 3). TNF-α for the three ungrouped cases had also significantly higher levels (43.3–61.8 pg/mL) compared to those registered for women with normal pregnancies.

IL-6 value for the PC1 differed slightly from that of the NPC set (47.5 ± 9.7 vs 30.6 ± 7 pg/mL) (Figure 3). The mean plasma level of IL-6 for the two patient groups EPL1 and EPL2 (99.4 ± 23 pg/mL and 179± 29 pg/mL, respectively), was much higher (*p* < 0.05) than that of the PC1 group (47.5 ± 9.7 pg/mL) (Figure 3). The values of IL-6 in the three ungrouped cases were statistically higher than of PC1 but lower than those of the EPL1 and EPL2 sets (Figure 3). The correlation between TNF-α and Il-6 levels in EPL1 and EPL2 sets was found positive (r = 0.74).

## 3. Discussion

Despite the significant advances in plasma proteomics in recent years, the discovery of reliable biomarkers for many pathologies, among which are the EPLs, remains a challenge for researchers. Therefore, in this work, we aimed at providing insight into the alterations occurring in the blood plasma proteins of EPL women.

For the first time, we examined the changes in the denaturation profiles of plasma proteins derived from women with EPL with unknown causes in relation to those of NPC and PC1 subjects. The presented results clearly demonstrate the altered thermal behavior of the plasma proteome of women with EPL, which is expressed in thermal stabilization of the albumin fraction with concomitant suppression of the albumin-assigned and the enhancement of the globulin-assigned transitions.

As pregnancy involves many changes in all aspects of the mother’s physiology, our first step was to determine if there is any alteration in plasma proteins’ thermal behavior of PC1 from that of NPC. The data revealed that the overall calorimetric profile of blood plasma from healthy pregnant women did not differ significantly from that of non-pregnant women. Since the PC1 sample was collected from women with a normal pregnancy in the first trimester, it was not surprising that no significant difference was detected in terms of calorimetric features of plasma proteins. In this regard, it should be noted that albumin, the most abundant protein in the plasma, decreases with the progress of pregnancy, nonetheless, in the first trimester, it is still within the reference values [44]. Immunoglobulins (IgG, IgA, and IgM) also decrease significantly in the second and third trimesters but remain almost unchanged in the first one [45]. Fibrinogen, which is involved in the coagulation process, is known to increase during pregnancy, leading to a hypercoagulable blood state [46]. Our results indeed demonstrate an increase in fibrinogen assigned transition in the PC1 samples. However, it should be noted, that the electrophoretic data concerning the fibrinogen plasma concentration are within the normal range. Thus, it can be assumed that the alteration in the Fg calorimetric transition is rather due to its altered binding/conformation state.

We have shown that more than half of the patients’ thermograms strongly differ from the control PC1 group. The most prominent feature for the EPL2 group was the stabilization of the main transition, which refers mainly to HSA denaturation, with a concomitant decrease in its amplitude and an upshift of the peak to ca. 64 °C. As a result, the entire thermogram is shifted to higher temperatures with an increase in both the enthalpy change and the weighted average center of the thermograms from 65.3 °C for the control group to 67.0 °C for the EPL2 group. Since serum electrophoresis data do not display changes in the albumin band for EPL samples (demonstrating that the HSA level is not altered), most likely, a fraction of the HSA is stabilized and hence the main transition is upshifted to higher temperatures. 

A unique feature in the denaturation curve of the EPL_case4_ was the strong destabilization of the main transition, along with reduced enthalpy change (ΔH) as compared to the PC1 group. No correlations were found between the calorimetric and electrophoretic data and the reason for the deviation from the reference values remains unclear. The plasma denaturation profiles for the two ungrouped cases EPL_case10_ and EPL_case12_, on the contrary, showed the signature of stabilization of the plasma proteome as compared to the PC1 group. The albumin assigned transition was upshifted by more than 5 °C, overlapped with the globulins peak, and thus a broad single melting transition was recorded in the range 65–75 °C. A peculiar feature of these cases was the enlarged IgG/transferrin-assigned transition above 80 °C. It should be noted that the β1-globulin fraction for these two ungrouped cases was above the reference values and thus most likely attributes to the elevated amplitude of the last transition at about 82–86 °C. The transition event at 82 °C could reflect the higher fraction of the CH3 domain of the Fc region of IgG [47]. The increased transferrin level in the high-temperature region could be another possible reason for the enhancement of this transition [48].

A different extent of HSA stabilization is commonly observed for cancer [37,49,50], and other diseases, but its precise origin is still obscure. Recently, we have applied a sophisticated mathematical approach (InterCriteria Analysis) to a large dataset of calorimetric and biochemical parameters derived for the serum proteome of patients with multiple myeloma that confirmed that the deviation of HSA assigned transition is not correlated with the protein concentration but is rather due to stabilization of a fraction of HSA [51]. Various pathology-related factors can affect the conformational state of HSA and thus its transition temperature, including protein and/or ligand binding as well as its oxidation status. It is conceivable that in general, the upshift of the main transition may result from the interaction of HSA with specific certain pathological state ligands. As mentioned, specific serum markers are characteristic of normal pregnancy. However, deviation from their normal values could lead to severe obstetric complications. Pregnancy-associated plasma protein-A (PAPP-A), low human chorionic gonadotropin (hCG), higher PAPP-A levels, and the level of inhibin A are associated with an increased frequency of adverse outcomes in the first trimester [27,52,53,54] and could be a marker for predicting miscarriage. The binding of some specific molecules to HSA, or other major plasma proteins that contribute to the observed thermal transitions, may partly explain the deviations of the patients’ thermograms from the control ones. Other factors, such as albumin oxidation, may also be related to the stabilization of the main transition. HSA plays a key part in the body’s antioxidant defense against reactive species and is the protein most exposed to ROS [55]. It has been already reported that oxidation enhances HSA thermal stability [56]. The study of Musante et al. also demonstrates a thermal stabilization in the in vivo condition of the albumin structure upon oxidation [57]. Oxidative stress is one of the risk factors for many pregnancy complications, including fetal loss [58]. Recently, Omeljaniuk et al. have shown that total antioxidant status and antioxidant enzymes such as glutathione peroxidase are significantly lower in women who suffered a miscarriage than in pregnant women at full-term delivery [59]. Oxidative stress is a key component of inflammatory reactions and is considered to be an important pathophysiologic process. Therefore, future detailed studies directed towards correlating the level of known pregnancy markers and oxidative stress in EPL patients, and the shift in HSA calorimetric transition are needed. 

In this work, we concentrate on the possible relation between pregnancy-related proinflammatory factors and the changes in the calorimetric features of the blood plasma of EPL patients.

It should be noted that the deviation of the EPL calorimetric plasma characteristics from the control ones does not correlate entirely with the changes in protein concentration determined from the electrophoresis data. Therefore, the observed calorimetric features are rather due to the changes in the conformation state of the major plasma proteins, attributed to chemical modification and/or intermolecular or ligand interactions.

The inflammatory process of the maternal–fetal relationship is critical to successful implantation and is largely regulated by cytokines [60]. In order to quantify the extent of the inflammatory process, we determined the levels of proinflammatory cytokines—TNF-α and IL-6 in the blood plasma of the studied groups of women as well as the carriage of 675 4G/4G polymorphism in the *PAI-1* thrombophilia gene. We found elevated values of TNF-α in EPL2 and all additional EPL cases, whereas IL-6 was elevated in both EPL groups as compared to the control groups. This effect most probably is an expression of an overall enhanced immune response in EPL women. 

Our data also reveal the significantly higher carriage of the 675 4G/4G polymorphism in the *PAI-1* gene in the EPL2 group and in two EPL cases (10 and 12) as compared to pregnant and non-pregnant control, which results in a higher level of *PAI-1*. It is important to be noted that *PAI-1* is considered an acute phase reactant, being closely influenced by inflammatory cytokines, (e.g., IL-6, interleukin-1, TNF-α), and growth factors secreted in the plasma. During a healthy pregnancy, the *PAI-1* level in the plasma gradually elevates reaching its maximum in the last month of pregnancy before the delivery. In early pregnancy, macrophages participating in the modulation of trophoblast invasion secrete TNF-α, which promotes extravillous cytotrophoblasts to release *PAI-1* during placentation, but its level in circulation is not raised in normal pregnancy [61]. However, in our study, we detected elevated levels of TNF-α in the EPL2 group, in addition to higher *PAI-1* levels because of 4G/4G carriage. We strongly suggested that these are among the trigger factors of unfavorable pregnancy outcomes in the EPL2 group. Recently it has been reported that the overexpression of *PAI-1* is an important indicator of pregnancy complications, including miscarriage [62]. 

Similarly to the data obtained from our analysis, previous studies described increased serum levels of TNF-α and IL-6 levels in patients with spontaneous abortion than in women with normal pregnancies [14,63,64,65]. The positive correlations which we found between TNF-α and Il-6 (the Th1 and Th2 cytokines) may be the expression of an overall raised immune response after pregnancy loss in the first trimester of pregnancy. Pro-inflammatory Th 1 cytokines (TNF-α) are necessary for stimulating vasculogenesis, which is essential for the implantation process in the early stage of normal pregnancy. Indeed, long-term exposure to Th1 cytokines can lead to a cell-mediated immune response that is harmful to the fetus and could result in miscarriage [65]. Stable maintenance of pregnancy requires a proper tuning/balance between Th1 and Th2 (IL-6) cytokines. Recently, it was observed that pregnancies with unfavorable outcomes are usually associated with an overall higher expression of Th1 cytokines whereas normal pregnancies are convoyed by a higher expression of Th2 cytokines [66]. Consequently, our research confirms that although both factors are raised in the EPL groups (the correlations between TNF-α and Il-6 in EPL groups were positive: r = 0.74), the increase in IL6 (Th2) is more pronounced than this in TNF-α. Although it is unlikely that these factors (TNF-α, Il-6, *PAI-1*) have a direct effect on the plasma calorimetric features, there is a clear correlation between their significantly higher levels in EPL 2 group and the considerable change in the respective thermograms. Extensive research is needed to explore the impact of proinflammatory and prothrombotic factors (TNF-α, Il-6, *PAI-1* and others) on the calorimetric features of blood plasma derived from women with adverse pregnancy outcomes, which may be helpful to develop novel diagnostic approaches to prevent the miscarriages in the future. The DSC approach might be helpful in this respect since it does not require the usage of expensive reagents and is a fast method. Therefore, future studies revealing the nature/origin of the abnormal calorimetric profiles in EPL patients might provide a useful micro-invasive diagnostic tool that can complement routine testing and be implemented in clinical practice. 

## 4. Materials and Methods

### 4.1. Selection of Patients and Healthy Controls

The study groups consisted of 26 patients (mean age 34 ± 8) diagnosed with EPL with unknown etiology, 18 healthy (mean age 36 ± 6) age-matched non-pregnant women (NPC group), and 8 pregnant women (mean age 31 ± 4) with uncomplicated first trimester (PC1 group). In order to ensure the reliability of the present research, several criteria were applied in the selection of volunteers included in the control groups. For the PC1 group, pregnant women without a history of previous miscarriages and without pathologies were recruited. We excluded the patients with genetic disorders, uterine anatomical abnormalities, hormonal abnormalities, (thyroid), concomitant infectious causes, antiphospholipid syndrome, immune disorders, and metabolic disorders such as diabetes or hypertension. For NPC, women with one or more live births, without any complications during and after pregnancy and childbirth were selected. Informed consent was obtained from all study participants. The study is approved by the Ethics Committee of Medical University Pleven (approval no. 404-KENID 22/10/15) and was performed in accordance with the Helsinki international ethical standards on human experimentation.

### 4.2. Blood Collection

For the EPL group, blood was collected prior to the administration of anesthesia in order to avoid its possible influence on the plasma samples. For the PC1 group, blood was collected during routine tests, and for NPC women this procedure was performed after morning fasting. Venous blood was collected in two 3 mL ethylene diamine-tetra acetic acid (EDTA) vacutainers (0.084 mL 15% EDTA Becton, Dickinson and Company, Franklin Lakes, NJ, USA). The blood from the first vacutainer was used for DNA analysis, and the one from the second vacutainer was used for plasma isolation.

### 4.3. Sample Preparation

Blood plasma was obtained after centrifugation of blood and further diluted in PBS buffer to the required concentration for DSC measurements.

### 4.4. Characterization of the Protein Content

The total protein content was determined by the Biuret method [67]. Capillary electrophoresis (Capillarys 2, Sebia, Lisses, France) was carried out to determine the levels of the main plasma proteins.

### 4.5. DSC Experiments

DSC thermograms were recorded in the temperature range of 30–95 °C with a scanning speed of 1 °C/min using a microcalorimetric system DASM-4 (Biopribor, Pushchino, Russia). The plasma solution in the calorimetric cell was reheated after the cooling from the first run to estimate the reversibility of the thermally induced transitions and the second scan was subtracted from the corresponding data of the first one. The obtained calorimetric curves were normalized to the plasma proteins’ concentration. The calorimetric data were evaluated using the OriginPro 2018 program package. The main thermodynamic parameters: specific heat capacity (c_P_^ex^) and transition temperature (T_m_) of the successive transitions, calorimetric enthalpy (ΔH), and mean weighted center of (T_FM_) of the thermograms were determined. 

### 4.6. DNA Analysis

The DNA for analysis was isolated from venous blood by salting out from non-frozen blood. The test of polymorphism in the *PAI-1* gene was accomplished by multiplex polymerase chain reaction (PCR) followed by hybridization with a diagnostic set strip assay (Viennalab, Vienna, Austria) according to the manufacturer’s protocol. Allele-specific PCR was used to detect 4G and 5G genotypes. Amplification was performed in two separate series, with the two constitutive primers and two inner primers corresponding to sequences 4G and 5G [68]. The DNA hybridization on a strip was based on the principle of selective amplification and detection of PCR products by immobilization on membrane carrier-specific probes.

### 4.7. ELISA for Cytokines 

The levels of TNF-α and IL-6 were determined by the enzyme-linked immunosorbent assay (ELISA) method [69]. Samples were tested in triplicate, and absorbance values were measured at 450 nm in an ELISA reader (Biosan, Latvia). Accurate concentrations of cytokines were determined by comparing their respective absorbencies with those obtained for the reference standards plotted on a standard curve using reference recombinant cytokines. 

### 4.8. Statistical Approaches

The statistical methodology developed by Fish et al. was applied to determine the degree of deviation of the patients’ thermograms from those of the respective controls [43]. This methodology determines the similarity metric ρ, which combines two statistical indicators—Pearson’s correlation coefficient, r, reflecting the similarity in shape of a test curve as compared to a set of control thermograms, and the spatial distance metric P, which takes into account the deviation of the distance of the test thermogram from the control thermogram for each experimental point. All data were expressed as means ± SD (standard deviation). The nonparametric Mann–Whitney test for independent samples was used to assess the statistical significance of differences between means. A *p*-value <0.05 was considered statistically significant.

## 5. Conclusions

The obtained results clearly demonstrate the altered thermal behavior of the plasma proteome of women with EPL (58% of cases). Thermal stabilization of a fraction of albumin-assigned transition with concomitant suppression of the major and enhancement of the globulin-assigned transition are characteristic features of EPL DSC profiles, that could be used as a new indicator of a risk pregnancy. The presented results suggest altered composition or intermolecular interactions of the plasma proteome for EPL women. 

In addition, the alterations of the EPL thermograms correlate with the prevalence of the polymorphism in *PAI-1* genes of thrombophilia and the increased levels of TNF-α and IL-6 established in the blood. Although it is unlikely that these factors (TNF-α, Il-6, *PAI-1*) have a direct effect on the plasma calorimetric features, there is a clear correlation between their significantly higher levels in EPL 2 group and the considerable change in the respective thermograms. 

The concomitant changes in plasma thermograms confirm the potential of the DSC approach for distinguishing changes in the pathological state of the blood plasma proteome.

Extensive research is necessary to explore the impact of proinflammatory and prothrombotic factors (TNF-α, Il-6, *PAI-1* and others) on the calorimetric features of blood plasma derived from women with adverse pregnancy outcomes, which may be helpful to develop novel diagnostic approaches to prevent the miscarriages in the future.

## Figures and Tables

**Figure 1 ijms-23-08764-f001:**
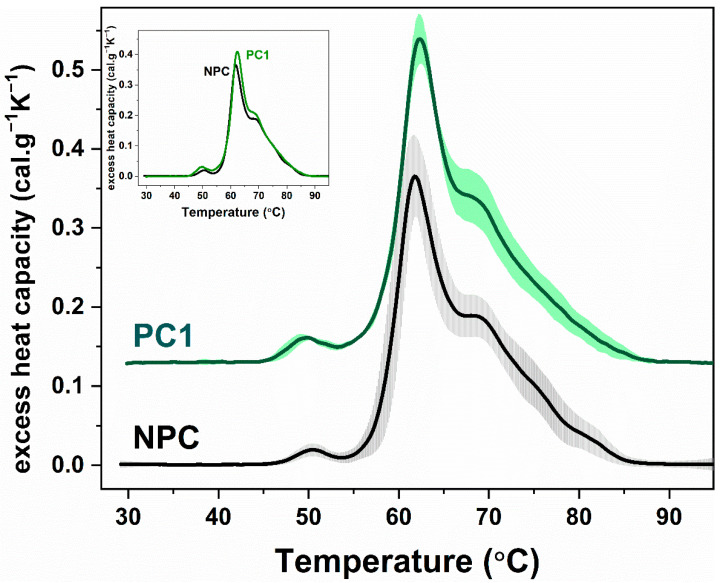
Average DSC profiles and standard deviation of the plasma proteome of control nonpregnant women (NPC, black solid line, gray shadow) and pregnant women registered in the first trimesters of pregnancy (PC1, panel A, dark green solid line, green shadow). For clarity, the thermograms are displaced vertically. Inset represents the overlapping average denaturation profiles of NPC and PC1 groups. All thermograms are recorded with a scan rate of 1 °C min^−1^ in the range of 30–95 °C.

**Figure 2 ijms-23-08764-f002:**
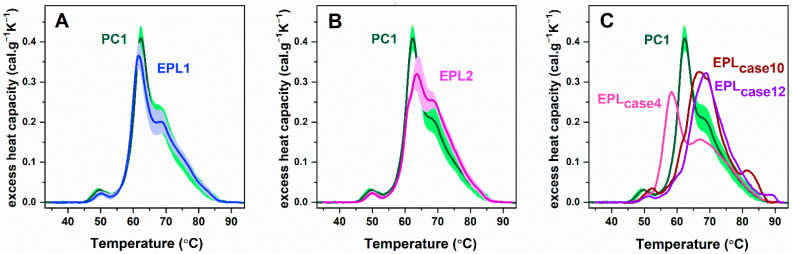
Calorimetric profiles (mean ± SD) of blood plasma derived from women with early pregnancy loss (EPL). Panel (**A**): EPL1 group (blue line, blue shadow); panel (**B**): EPL2 group (magenta line, light violet shadow); panel (**C**): ungrouped thermograms of cases 4 (pink line), 10 (wine line), and 12 (violet line). For clarity, the average blood plasma profile registered for healthy pregnant women (PC1, dark green line, and green shadow) is presented in each panel. All thermograms are recorded with a scan rate of 1 °C min^−1^ in the range of 30–95 °C.

**Figure 3 ijms-23-08764-f003:**
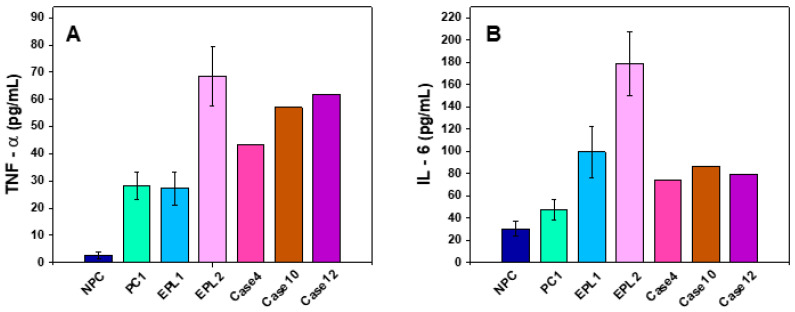
TNF-α (**A**) and IL-6 (**B**) cytokine levels determined for non-pregnant women (NPC), pregnant women in the first trimester of pregnancy (PC1), and for patients with early pregnancy loss (EPL).

**Table 1 ijms-23-08764-t001:** Main patients’ characteristics and biochemical parameters (total protein (TP), human serum albumin (HSA), C-reactive protein (CRP), fibrinogen (Fg) level), determined for non-pregnant (NPC) and pregnant women in the first trimester of pregnancy (PC1), and patients with early pregnancy loss (EPL). Mean values ± SD.

Groups (№ of Cases)	Age (Years)	GW	BMI (kg/m^2^)	TP (g/L)	HSA (g/L)	CRP (mg/L)	Fg (g/L)
NPC (n = 18)	36 ± 6	-	20.1 ± 1.1	72.6 ± 2.7	46.8 ± 1.9	3.4 ± 1.1	3.2 ± 0.6
PC1 (n = 8)	31 ± 4	7 ÷ 12	21.6 ± 1.3	69.7 ± 2.9	45.9 ± 1.7	5.8 ± 0.9 **	3.5 ± 0.8
EPL (n = 26)	34 ± 8	6 ÷ 12	22.4 ± 3.6	70.0 ± 3.2	45.9 ± 1.7	3.3 ± 0.6 *	3.7 ± 0.7

* Indicates statistically significant difference (*p* < 0.05) from the respective PC1 control values; ** Indicates statistically significant difference (*p* < 0.05) from the NPC values.

**Table 2 ijms-23-08764-t002:** Thermodynamic parameters (temperatures of denaturation, Tm, and excess heat capacities of the successive thermal transitions, c_P_; the c_P_ ratio of albumin and immunoglobulins assigned transitions, c_P_^HSA^/c_P_^Igs^; calorimetric enthalpy, ΔH, and weighted average center of the thermogram, T_FM_), estimated from the DSC profiles of blood plasma for non-pregnant (NPC) and pregnant women in the first trimester of pregnancy (PC1).

Groups	c_P_^Fg^ (cal·g^−1^·K^−1^)	T_m_^HSA^ (°C)	c_P_^HSA^ (cal·g^−1^·K^−1^)	T_m_^Igs^ (°C)	c_P_^Igs^ (cal·g^−1^·K^−1^)	c_P_^HSA^/c_P_^Igs^	ΔH (cal·g^−1^)	T_FMs_ (°C)
NPC	0.019 ± 0.008	61.6 ± 0.4	0.37 ± 0.03	68.4 ± 0.7	0.19 ± 0.04	2.00 ± 0.3	4.2 ± 0.5	65.2 ± 0.4
PC1	0.031 ± 0.003 *	62.2 ± 0.4	0.41 ± 0.04	68.5 ± 0.5	0.2 ± 0.03	2.05 ± 0.4	4.6 ± 0.3	64.7 ± 0.6

* Indicates statistically significant difference (*p* < 0.05) from the respective NPC control value.

**Table 3 ijms-23-08764-t003:** Thermodynamic parameters: denaturation temperatures (T_m_) and excess heat capacities (c_P_) of the main thermal transitions; the c_P_ ratio of albumin and globulins assigned transitions, c_P_^HSA^/c_P_^Igs^; calorimetric enthalpy (ΔH); weighted average center of the thermograms (T_FM_), and similarity parameters (r, P, ρ according to Fish et al. 2010), estimated from the DSC profiles of blood plasma for pregnant women in the first trimester of pregnancy (PC1) and for patients with early pregnancy loss (EPL). The data are presented as mean values ± SD.

Groups (№ of Cases)	c_P_^Fg^ (cal·g^−1^·K^−1^)	T_m_^HSA^ (°C)	c_P_^HSA^ (cal·g^−1^·K^−1^)	T_m_^Igs^ (°C)	c_P_^Igs^ (cal·g^−1^·K^−1^)	c_P_^HSA^/c_P_^Igs^	ΔH (cal·g^−1^)	T_FM_ (°C)	r	P	ρ
PC1 (n = 8)	0.031 ± 0.003	62.2 ± 0.4	0.41 ± 0.04	68.5 ± 0.5	0.2 ± 0.03	2.05 ± 0.4	4.6 ± 0.3	64.7 ± 0.6			
EPL1 (n = 11)	0.020 ± 0.005 *	61.5 ± 0.1	0.37 ± 0.02 *	68.5 ± 0.4	0.20 ± 0.03	1.85 ± 0.2	4.5 ± 0.3	65.3 ± 0.3	0.94	0.82	0.81
EPL2 (n = 12)	0.023 ± 0.004	63.6 ± 0.2 *	0.31 ± 0.04 *	68.5 ± 0.3	0.25 ± 0.03 *	1.24 ± 0.3 *	4.7 ± 0.1 *	67.0 ± 0.2 *	0.86	0.67	0.70
Ungrouped:											
EPL_case4_	0.021	58.3 *	0.27 *	67.1 *	0.15 *	1.75 *	4.1 *	64.1	0.63	0.89	0.81
EPL_case10_	0.035	66.7 *	0.32	69.3	0.31	n.d.	4.8 *	68.3 *	0.34	0.86	0.68
EPL_case12_	0.014	69.0 *	n.d.	68.9	0.32	n.d.	4.3	69.6 *	0.26	0.81	0.52

* Indicates statistically significant difference (*p* < 0.05) from the respective PC1 control value; n.d. –not determined.

**Table 4 ijms-23-08764-t004:** Concentration of the main plasma proteins fractions (mean ± SD) determined by capillary electrophoresis and presented as a percentage of the total protein content for the groups under study.

Groups	HSA (%)	α1 (%)	α2 (%)	β1 (%)	β2 (%)	γ (%)
Reference values	54.7–69.66	2.63–5.03	4.87–10.48	5.35–9.19	2.38–7.11	9.69–18.9
PC1	56.6 ± 3.2	5.2 ± 0.5	11.0 ± 0.2 *	8.6 ± 0.8	5.6 ± 0.7	12.3 ± 0.7
EPL1	57.9 ± 3.4	5.8 ± 0.9 *	10.0 ± 1.0	7.0 ± 0.6	7.5 ± 0.6	11.7 ± 2.8
EPL2	59.0 ± 3.1	6.0 ± 0.7 *	9.1 ± 1.1	7.2 ± 0.5	8.2 ± 0.9 *	12.7 ± 2.1
Ungrouped						
EPL_case4_	51.2	6.0 *	8.9	8.3	6.9	18.7
EPL_case10_	62.1	5.6 *	4.5	9.3 *	8.6 *	9.9
EPL_case12_	57.3	5.3 *	8.9	9.7 *	8.4 *	10.4

* Indicates statistically significant difference (*p* < 0.05) from the respective PC1 control values.

**Table 5 ijms-23-08764-t005:** Statistical data on the prevalence, odds ratio, 95% confidence interval, chi-squared, and *p*-value of 675 4G/4G (*PAI-1*) genotype carriage in EPL1 and EPL2 groups versus healthy pregnant and non-pregnant controls (NPC + PC).

Groups	4G/4G (*PAI-1*) Carriers (%)	OR	95% CI	CHI Squared	*p*-Value
EPL2	58.3	6.4400	1.4358–28.885	6.5533	0.010469
EPL1	27.2	1.0222	0.1670–6.2581	0.056	0.9810
NPC + PC1	19.2				

% Denotes the percentage of cases carrier of *PAI-1* polymorphism from the total number of cases in the respective group.

## Data Availability

The data are contained within the article.

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
