# Peer review of "Altered Thermal Behavior of Blood Plasma Proteome Related to Inflammatory Cytokines in Early Pregnancy Loss"

_ijms, 2022, doi:10.3390/ijms23158764_

Round 1
Reviewer 1 Report
Comments to the authors:
In this manuscript, authors reported that “Altered thermal behavior of blood plasma proteome related to 2 inflammatory cytokines in early pregnancy loss”. The authors suggested that 58% of the EPL thermograms differ significantly from those of healthy pregnant women. Thermal stabilization of a fraction of albumin-as-signed transition with a concomitant suppression of the major and enhancement of the globulin-assigned transition are characteristic features of EPL calorimetric profiles that could be used as a new indicator of a risk pregnancy. The authors also concluded that an altered composition or intermolecular interactions of the plasma proteome of women with EPL. The authors pointed out that the alterations of the EPL thermograms correlate with the increased blood levels of TNF-α and IL-6 and a higher prevalence of the polymorphism in PAI-1 gene, suggesting an expression of an overall enhanced immune response. These results indicate that the potential of the DSC approach for distinguishing changes in the pathological state of the blood plasma proteome. Therefore, the information presented in this manuscript is not similar to previous studies. However, there are some interesting points in this manuscript to be revealed if the authors can address some critical questions as below:
General comments:
1. In the results, there are some controversial data regarding the levels of proinflammatory cytokines TNF-α and IL-6 during pregnancy and its complications. This point should be adequately addressed.
2. Although it is unlikely that these factors (TNF-α, Il-6, PAI-1) have a direct effect on the plasma calorimetric features, there is a clear correlation between their significantly higher levels in EPL 2 group and the considerable change in the respective thermograms. This point was already adequately addressed.
3. The extensive research is necessary to explore the impact of proinflammatory and prothrombotic factors (TNF-α, Il-6, PAI-1and others) on the calorimetric features of blood plasma derived from women with adverse pregnancy outcomes, which may be helpful to develop novel diagnostic approaches to prevent the miscarriages in the future.
4. The current version of the manuscript raises few issues that need to be addressed to strengthen the manuscript and enhance its general interest and significance to be acceptable.
Author Response
Comments to the authors:
In this manuscript, authors reported that “Altered thermal behavior of blood plasma proteome related to 2 inflammatory cytokines in early pregnancy loss”. The authors suggested that 58% of the EPL thermograms differ significantly from those of healthy pregnant women. Thermal stabilization of a fraction of albumin-as-signed transition with a concomitant suppression of the major and enhancement of the globulin-assigned transition are characteristic features of EPL calorimetric profiles that could be used as a new indicator of a risk pregnancy. The authors also concluded that an altered composition or intermolecular interactions of the plasma proteome of women with EPL. The authors pointed out that the alterations of the EPL thermograms correlate with the increased blood levels of TNF-α and IL-6 and a higher prevalence of the polymorphism in PAI-1 gene, suggesting an expression of an overall enhanced immune response. These results indicate that the potential of the DSC approach for distinguishing changes in the pathological state of the blood plasma proteome. Therefore, the information presented in this manuscript is not similar to previous studies. However, there are some interesting points in this manuscript to be revealed if the authors can address some critical questions as below:
Dear Reviewer,
We would like to express our thanks to the reviewers for the substantial comments and suggestions. By accepting most of them we believe it has resulted in an improved version of the manuscript. We enclose the Point-by-point response to reviewers’ comments for your consideration.
General comments:
- In the results, there are some controversial data regarding the levels of proinflammatory cytokines TNF-α and IL-6 during pregnancy and its complications. This point should be adequately addressed.
Answer: We thank the reviewer for the noted controversy. It was a technical error and not a clear result description. Now it was corrected. Page 7, Lines 251.
- Although it is unlikely that these factors (TNF-α, Il-6, PAI-1) have a direct effect on the plasma calorimetric features, there is a clear correlation between their significantly higher levels in EPL 2 group and the considerable change in the respective thermograms. This point was already adequately addressed.
Answer: We appreciate this comment of the reviewer.
- The extensive research is necessary to explore the impact of proinflammatory and prothrombotic factors (TNF-α, Il-6, PAI-1and others) on the calorimetric features of blood plasma derived from women with adverse pregnancy outcomes, which may be helpful to develop novel diagnostic approaches to prevent the miscarriages in the future.
Answer: We thank the reviewer for this suggestion. The aforementioned text was appropriately corrected in the text. Page 10 (the last paragraph), lines 615-618.
- The current version of the manuscript raises few issues that need to be addressed to strengthen the manuscript and enhance its general interest and significance to be acceptable.
Answer: According to the reviewer's advice along with the above-addressed points, we emphasized the important finding of the investigation, and the following text was added on Page 8 (3. "Discussion" section; the second paragraph), lines 273-276:
The presented results clearly demonstrate the altered thermal behavior of the plasma proteome of women with EPL, which is expressed in thermal stabilization of the albumin fraction with concomitant suppression of the albumin-assigned and the enhancement of the globulin-assigned transitions.

Reviewer 2 Report
I would like to thank the Editor for giving me the chance to read the “Altered thermal behavior of blood plasma proteome related to 2 inflammatory cytokines in early pregnancy loss”. In their work authors aim at assessing the thermodynamic behavior of blood plasma/serum proteome in recurrent pregnancy loss (RPL) women as compared to healthy pregnant women. They found that 58% of the EPL thermograms differed significantly from those of healthy pregnant women.
The paper has several limitations. Therefore, I would not recommend it for publication in the Internation Journal of Molecular Sciences.
In spite of the originality of the research, the impact of the results is too weak. They obtained results by comparing RPL women with pregnant women which represent too different populations. Also in their discussion they include a lot of mechanisms hence resulting a little confusing.
Blood parameters which are not specific of an ongoing pregnancy
body mass index has not been included in the patients characteristics
no data about possible concomitant infections are provided
In the discussion authors cite a lot of mechanisms hence resulting a little confusing.
It is difficult to appreciate the clinical implications of the obtained results.
Author Response
Dear Reviewers,
We would like to express our thanks to the reviewers for the substantial comments and suggestions. By accepting most of them we believe it has resulted in an improved version of the manuscript. We enclose the Point-by-point response to the reviewers’ comments for your consideration.
Reviewer 2
|
|
Yes |
Can be improved |
Must be improved |
Not applicable |
|
|
Does the introduction provide sufficient background and include all relevant references? |
(x) |
( ) |
( ) |
( ) |
|
|
Are all the cited references relevant to the research? |
( ) |
(x) |
( ) |
( ) |
|
|
Is the research design appropriate? |
( ) |
( ) |
(x) |
( ) |
|
|
Are the methods adequately described? |
( ) |
( ) |
(x) |
( ) |
|
|
Are the results clearly presented? |
( ) |
( ) |
(x) |
( ) |
|
|
Are the conclusions supported by the results? |
( ) |
( ) |
(x) |
( ) |
|
Comments and Suggestions for Authors
I would like to thank the Editor for giving me the chance to read the “Altered thermal behavior of blood plasma proteome related to 2 inflammatory cytokines in early pregnancy loss”. In their work authors aim at assessing the thermodynamic behavior of blood plasma/serum proteome in recurrent pregnancy loss (RPL) women as compared to healthy pregnant women. They found that 58% of the EPL thermograms differed significantly from those of healthy pregnant women.
The paper has several limitations. Therefore, I would not recommend it for publication in the Internation Journal of Molecular Sciences.
In spite of the originality of the research, the impact of the results is too weak.
Answer: We thank the reviewer for the substantial comments.
Please kindly find our answers point-by-point.
1. They obtained results by comparing RPL women with pregnant women which represent too different populations.
Answer: In the investigation, only women with early pregnancy loss are included, but not with a recurrent pregnancy loss. Both groups were idiopathic.
2. Also in their discussion they include a lot of mechanisms hence resulting a little confusing.
Answer: The discussion part of the manuscript with the suggested mechanisms was significantly reduced,
- Blood parameters which are not specific of an ongoing pregnancy .
Answer: We thank the reviewer for this comment. All hematological parameters were studied for the control groups and the EPL group of women. In Table 1, we have listed only those parameters that are directly related to blood plasma. The remaining parameters related to erythrocytes and platelets (RBC, Hb, Ht, HCT, MCV, MCH, MCHC, and Platelet count) are included in our previous studies [Langari A., Danailova A., Krumova S., Komsa-Penkova R., Golemanov G., Giosheva I., Gartchev E., Taneva S.G., Todinova S. Aging-related changes in the calorimetric profile of red blood cells from women with miscarriages. Journal of Thermal Analysis and Calorimetry volume, 2020, 142, 1919-1926. https://doi.org/10.1007/s10973-020-10112-3; Andreeva T., Komsa-Penkova R., Langari A., Krumova S., Golemanov G., Georgieva G.B., Taneva S.G., Giosheva I., Mihaylova N., Tchorbanov A., Todinova S. Morphometric and Nanomechanical Features of Platelets from Women with Early Pregnancy Loss Provide New Evidence of the Impact of Inherited Thrombophilia. Int. J. Mol. Sci., 2021, 22, 7778. DOI: https://doi.org/10.3390/ijms22157778]. Therefore, we discuss parameters that might influence the calorimetric scans of blood plasma.
The patients with idiopathic EPL only were included in the study aiming to find a new potential marker. The usual lab investigations were in the reference range. Correspondingly we updated the gap in the description of the selected patients. Page 11 (4.1. "Selection of patients and healthy controls"; first paragraph), lines 794-796:
We excluded women with genetic disorders, uterine anatomical abnormalities, hormonal abnormalities, (thyroid), concomitant infectious causes, antiphospholipid syndrome, immune disorders, and metabolic disorders like diabetes or hypertension.
The patients with idiopathic EPL only were included in the study.
4. Body mass index has not been included in the patients characteristics
Answer: BMI is now included in the patients’ characteristics (Table 1, p. 3)
- no data about possible concomitant infections are provided
Answer: The studied patients did not have concomitant infections. This is now clarified in the patient’s selection criteria. Please concern the answer to the Comment 3
6. In the discussion authors cite a lot of mechanisms hence resulting a little confusing.
Answer: Please concern the answer to Comment 2. We corrected accordingly the Discussion in order to make it clearer. We also identify several directions for future studies that will complement our work and will further shed light on the nature of the observed thermal stabilization of albumin and the serum proteome as a whole in EPL patients. In this work, we explore the correlation of the observed calorimetric changes with the level of pro-inflammatory factors, namely the cytokines TNF-α and IL-6, as well as the carriage of 675 4G/4G polymorphism in the PAI-1 thrombophilia gene.
7. It is difficult to appreciate the clinical implications of the obtained results.
Answer: We have taken this comment into consideration, we already mentioned it in the revised manuscript. We clarified the clinical implication of the obtained results (p. 10):
Extensive research is necessary to explore the impact of proinflammatory and prothrombotic factors (TNF-α, Il-6, PAI-1and others) on the calorimetric features of blood plasma derived from women with adverse pregnancy outcomes, which may be helpful to develop novel diagnostic approaches to prevent the miscarriages in the future. The DSC approach might be helpful in this respect since it does not require the usage of expensive reagents and is a fast method. Therefore, future studies revealing the nature/origin of the abnormal calorimetric profiles in EPL patients might provide a useful micro-invasive diagnostic tool that can complement routine testing and be implemented in clinical practice.

Reviewer 3 Report
The paper is overall really interesting, materials and methods and results are clearly presented. However, material and methods should be section “2” , results section “3” and then the discussion. Please, move the results section after the introduction and the others in the order mentioned.
Author Response
Dear Reviewer,
We would like to express our thanks to the reviewers for the substantial comments and suggestions. By accepting most of them we believe it has resulted in an improved version of the manuscript. We enclose the Point-by-point response to the reviewers’ comments for your consideration.
Reviewer 3
|
|
Yes |
Can be improved |
Must be improved |
Not applicable |
|
|
Does the introduction provide sufficient background and include all relevant references? |
(x) |
( ) |
( ) |
( ) |
|
|
Are all the cited references relevant to the research? |
(x) |
( ) |
( ) |
( ) |
|
|
Is the research design appropriate? |
(x) |
( ) |
( ) |
( ) |
|
|
Are the methods adequately described? |
(x) |
( ) |
( ) |
( ) |
|
|
Are the results clearly presented? |
(x) |
( ) |
( ) |
( ) |
|
|
Are the conclusions supported by the results? |
(x) |
( ) |
( ) |
( ) |
|
Comments and Suggestions for Authors
The paper is overall really interesting, materials and methods and results are clearly presented. However, material and methods should be section “2” , results section “3” and then the discussion. Please, move the results section after the introduction and the others in the order mentioned.
Answer: We appreciate this important for us evaluation of our investigation. We thank the reviewer for the positive comments. We would like to point out that according to the Author's instructions provided by Int. J. Mol. Sci. the Research manuscript sections are: Introduction, Results, Discussion, Materials and Methods, and Conclusions (optional). Therefore, we kept the structuring of the paper in the present condition.
